# Effect of the Spatially-Varied Electron Mean Free Path on Vortex Matter in a Superconducting Pb Island Grown on Si (111)

Jesús González [1,*], Jader González [2], Fernando Durán [2], Carlos Salas [1] and Jorge Gómez [1]

1 Faculty of Engineering del Magdalena, Universidad del Magdalena, Carrera 32 No 22-08, Santa Marta 470004, Magdalena, Colombia; carlos.salas@unad.edu.co (C.S.); jgomez@unimagdalena.edu.co (J.G.)
2 Faculty of Engineering, Universidad Pontificia Bolivariana-Bucaramanga, Autopista Piedecuesta Kilómetro 7, Floridablanca 681004, Colombia; jader.gonzalez@upb.edu.co (J.G.); fernando.duran@upb.edu.co (F.D.)
* Correspondence: jgonzaleza@unimagdalena.edu.co

**Abstract:** In this work we report theoretical calculations of a superconducting island in a strong vortex confinement regime. The obtained results reveal the evolution of the superconducting condensate with an applied magnetic field, depending on the spatial profile of the electron mean-free path in the sample. The results of this study provide an insight about the emergent superconducting properties under such conditions, using the Ginzburg-Landau numerical simulations where spatial variation of thickness of the island and the corresponding variation of the mean free path, omnipresent in similar structures of Pb grown on Si (111), are taken into account. These results offer a new route to tailor superconducting circuits by nanoengineered mean free path, using for example the controlled ion-bombardment on thin films, benefiting from the here shown impact of the spatially-varying mean free path on the vortex distribution, phase of superconducting order parameter, and the critical fields.

**Keywords:** strong vortex confinement; superconductivity; Ginzburg–Landau theory; superconducting island; free energy





## 1. Introduction

Superconductivity is an electronic state of matter characterized by specific length scales: the coherence length $\xi$, the length scale of the Cooper-paired electrons, and the London penetration length $\lambda$, the length scale of magnetic field penetration into the superconductor [1]. The superconducting state is attained below certain critical values of temperature, magnetic field, and applied current, and the resistance to the passage of an electric current exhibited by the material is zero. The behaviour of the superconducting state can change dramatically when the size of a superconductor becomes comparable to the characteristic lengths of superconductivity, the coherence length $\xi$ and penetration depth $\lambda$ [2–5], or the sample is exposed to excitations on submicron scale [6,7]. In the past two decades several theoretical studies focused on the confinement effects in superconductors patterned on a scale lower than one or both of these characteristic scales [8–10], but also, focused on the effect of symmetry on condensate confinement comparing mesoscopic disks, squares and triangles [11–14]. The theoretical macroscopic framework that built upon the definition of a superconducting wavefunction that characterizes the superconducting state, comes from the Ginzburg–Landau theory [2,3] which is used in this work.

Most experimental focused on measuring the overall response of previous mentioned systems and superconducting islands [15–19], or to the local study of the magnetic confinement of vortices considering low magnetizing field $H \ll H_{c2}$ on large samples $d \sim \lambda \gg \xi$ [20,21]. The vortex configuration in real space at high fields $H \gg H_{c1}$ was studied in [17], but low confinement conditions were still considered, $d \gg \xi$. Nowadays, penetration and expulsion of vortices in mesoscopic superconducting samples has been

studied in Ref. [22–28], but the detailed picture of a strongly confined superconducting state $d \sim \xi \ll \lambda_{eff}$ remained incomplete, especially in samples of varying thickness and correspondingly changing electron mean-free path $l$. The simulations to change the latter unfavorable picture form the core objective of this paper, and will be demonstrated on the example of typical crystalline Pb islands grown on Si (111) (see Figure 1).

The outlook of the study is broader, since there exist several other means to tailor experimentally the mean-free path in superconducting samples. For example Focused Ion Beam Induced Deposition (FIBID) and Focused Electron Beam Induced Deposition (FEBID) are two very similar nanopatterning techniques that use of a focused beam of charged particles, either ions or electrons [29–31], and can be used to controllably induce disorder in superconducting films and thereby vary the mean-free path. In general, irradiation-induced doping or disorder exhibits higher accuracy compared to other methods, such as substitutional chemistry, in which there are greater uncertainties and inhomogeneity in phase distribution in the induced disorder [32]. Irradiation of superconductors is readily employed to significantly increase the normal-state resistivity and tailor (suppress) $Tc$, as well as to enhance the critical current density when efficient pinning centers are created [33,34].

One thus expects that controlled patterning of the mean-free path in superconductors [35] can drive nanofabrication techniques further, because there is a perennial need for both industry and research to exploit such fundamental effects for the development of quantum technologies [36], in which superconducting nanodevices represent a major field of research. Such devices include magnetic sensors in the form of superconducting interference superconducting quantum interference devices (SQUIDs) [37], single-photon detectors [38], quantum bits [39] and quantum switches [40], to name a few.

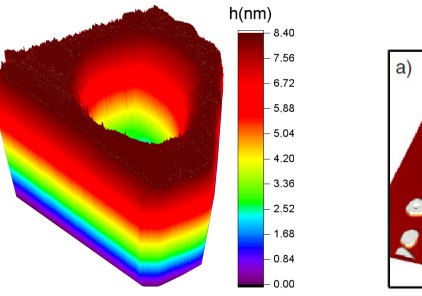
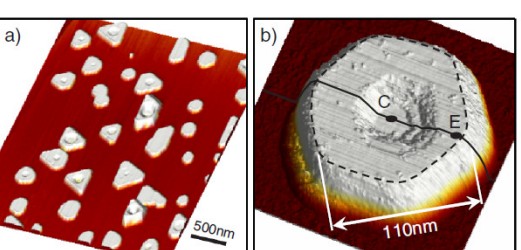

**Figure 1.** (Color online) (**Left**) Scanning tunneling microscopy (STM) images of Pb islands on Si (111) surface taken on a large scale (**a**) and on a local scale (**b**), where locations C and E correspond, respectively, to the island center and edge (reproduced with permission from [22]). The panel on the right shows the superconducting island used in simulations, made to mimic the characteristic shapes of Pb islands shown in (**a**). The taken lateral extent of the island is ≈4.6 nm, and its thickness in the border $d \approx 8.4$ nm, whereas inside the hole $d \approx 2.52$ nm, corresponding to 2–3 single atomic layers of Pb in (111) direction.

## 2. Results and Discussion

The process started calculating the full free-energy ($F$) spectrum and the corresponding vortex states as a function of applied magnetic field ($H_0$) for the island with a central hole and $l_\xi = 1.0$ Figure 2. Initially, the magnetic field was kept perpendicular to the plane bottom of Pb island. The method for finding vortex states is multifold: First, the applied magnetic field is increased and decreased in the considered range with a kept history of the previously found states in the field sweep; Second, considering each value of the magnetic field, the calculation is initialized from the fluctuating normal state (randomized $|\Psi| < 0.01$) and from the fully superconducting state ($|\Psi| \approx 1$). Following these steps, we construct the energy diagram of all stable vortex states, including those with higher energy. As a result, we found more than one stable vortex distribution, i.e., one with the lowest energy (ground state) and several others with higher energy (usually called metastable).

We show lower energies and higher stabilities represented in the wide ranges of magnetic fields in which the same number of vortices remains (see Figure 2, cases a., b., c., d. and e.). This finding demonstrates the strong confinement imposed on vortices by Meissner currents on the edge of the island (see Figure 1), where the Meissner current is strongest, as we can notice in the snapshots of superconducting current (see Figure 2 (left) inset) and it confines vortices toward the interior of the sample. On the other hand, it can also be noted that stability decreases with the entry of a greater number of vortices when the magnetic field is increasing (see Figure 2, cases f., h., and g.), due to vortices being basically parallel magnetic moments that destroy the superconductivity with an increase of their number. Additionally, other evidence of the strong confinement on the superconducting island can be noticed that vortices repel each other, and this repulsion, in the absence of sample boundaries, leads to the formation of a triangular (Abrikosov) lattice, which is not the case in this research.

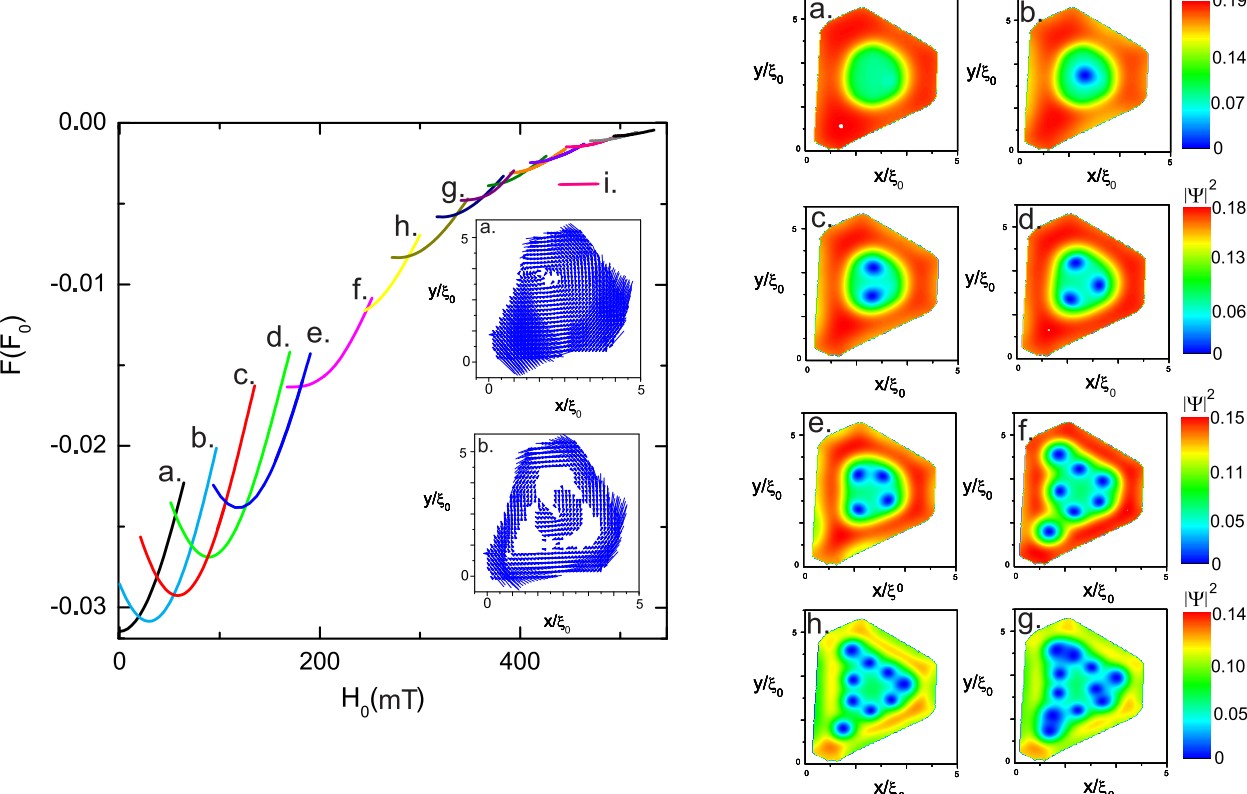

**Figure 2.** (Color online) (**Left**) Free energy curves and corresponding vortex states on a Pb island with a central hole demarcated with the letters a. until g., where the intensity of superconducting condensate is relate to the color bar of the islands snapshot evolves from red to blue, reflecting the reduction of the superconducting condensate strength. Those states show the range of the magnetic field for which the number of vortices remains stable within the sample i.e. for curve a. (black) there are no vortices ($L = 0$), as can be seen in snapshot a. of the superconducting condensate in the first column, located to the right. In this way, we can see the correspondence of the letter assigned to the curve and the snapshots of the vortices confined by the superconducting island, therefore, curve b. (blue) there is a vortex ($L = 1$), the curve c. (red) two vortices ($L = 1$) and in this way, we analyse the magnetic field ranges with the stable states of vortices. The snapshots within the Gibbs free energy graph show the intensity of the superconducting current, where the intensity of the current is evidenced by the intensity of the blue color in the sample, while the white areas show the areas in the normal state. The snapshots a. and b. of the superconducting current correspond to the instantaneous a. and b. of the superconducting condensate.

Figure 3a–d inset show some states where vortices have reached the center of the superconductor island, which can be verified by the circular diagram phase of the order parameter on the top of the island simulation. The vortex state is characterized by the total angular momentum $L$ through $\Psi = \psi exp(iL\phi)$, but, we can introduce an analog to the total angular momentum which is still a good quantum number. Choosing circular loops at the periphery of the center of the island, we find that the effective angular momentum $L = \Delta\phi/2\pi$, and each time that in the path the color went from red to blue a vortex is found $L = 1, 2, 3, ....$ The superconducting order parameter $|\Psi|$, nucleated at the sample surface, traps them inside in the island hole, carrying flux $L\phi_0$ where $\phi_0$ is the quantum flux. To check this "flux compression" model quantitatively, the self-consistent solution of the full Ginzburg–Landau (GL) equations is necessary Equation (3).

The transition between the two quantum states $L = 0$ (zero vortex) and $L = 1$ (one vortex) can be used to calculate the field $H_1$, corresponding to the penetration of the first flux line into a superconducting island. One option to find the value of $H_0$ in which these transitions take place is considering every curve of Figure 3a–d where the peaks obtained at a lower value of applied magnetic fields in every curve of Figure 3 ($F$ versus $H_0$), represent the first vortex entrance in the superconducting island, we can notice that it take place at different values of $H_0$. Figure 3 also can be used to calculate the field $H_0$ corresponding to the penetration of one flux line at different electron mean-free path $l_\xi$, and thus the transition between the two quantum states $L = 0$ and $L = 1$, as well as by using $l$ we can tune the number of peaks or transitions.

The characteristics lengths of a superconductor, the coherence length, and the penetration depth, take the effective values $\lambda_{eff} = 0.65\lambda_0\sqrt{\xi_0/l}$ and $\xi_{eff} = 0.85\sqrt{l\xi_0}$ considering $l_\xi = l_0/\xi_0$. Thus, the situation in the island is similar to that of a type II superconductor in the diffusive limit, in which correlation length $\xi$ and penetration depth $\lambda$ in a magnetic field depends on $l^{0.5}$ and $l^{-0.5}$, respectively. Therefore, the magnetic field in a superconductor is affected due to the increment of $\lambda$ with the increases of concentration of impurities, but also a strong variation of the number of superconducting electrons, i.e., electrons linked in Cooper pairs with the decrease of $\xi$. This behavior is shown in Figures 2 and 3. (inset) where variations of condensate are reached considering a spatial change of the electron mean free path.

In addition, the order parameter $|\Psi_0|^2$ is modified through the sample with the modification of $l$, as we expected, according to the proportionality between the order parameter and penetration depth which is $|\Psi_0|^2 \sim 1/\lambda$. It implies changes in strong screening supercurrents (see Figure 2 inset) that may circulate in the island, where strong supercurrent carry on a strong diamagnetic (Meissner) effect as we can notice in Figure 3 where the $H_1$ (first vortex entry) is reached (see also Table A1 Appendix A). It also can be noticed in the modification of $l$ at different zones of the sample (see Figure A1 Appendix B), which allows choosing not only the location of the vortex entrance but also the value of $H_0$.

Figure 4(left) shows the proportionality of the green curve corresponding to the values of the second critical magnetic field where the normal state is reached ($H_2$) versus the mean free path ($l$), which shows a result not reported before for this type of heterostructures. The fine-tuning of $H_2$ shows great applications for the design of electronic devices, through the bombardment of the Pb islands, which can be controlled using high precision with current technologies. Thus, it is clear that $H_1$ can be tuned, which supposes a variation in the Meissner current and with it the repulsion of the applied magnetic field $H_0$ and entry of vortices in the superconducting island. We know that flux penetration in type-II superconductors occurs in the form of quantized, flux-enclosing, supercurrent vortices. However, a more detailed analysis shows that this flux penetration must first overcome an energy barrier at the surface, which is called the Bean-Livinstong barrier (BLB). Corresponding variations of the Gibbs free energy with $l$, for several values of $H_0$, are shown in Figure 3. One can see that $H_1$ takes different values due to the fact that, in order to penetrate the superconductor island, the vortex must first overcome BLB.

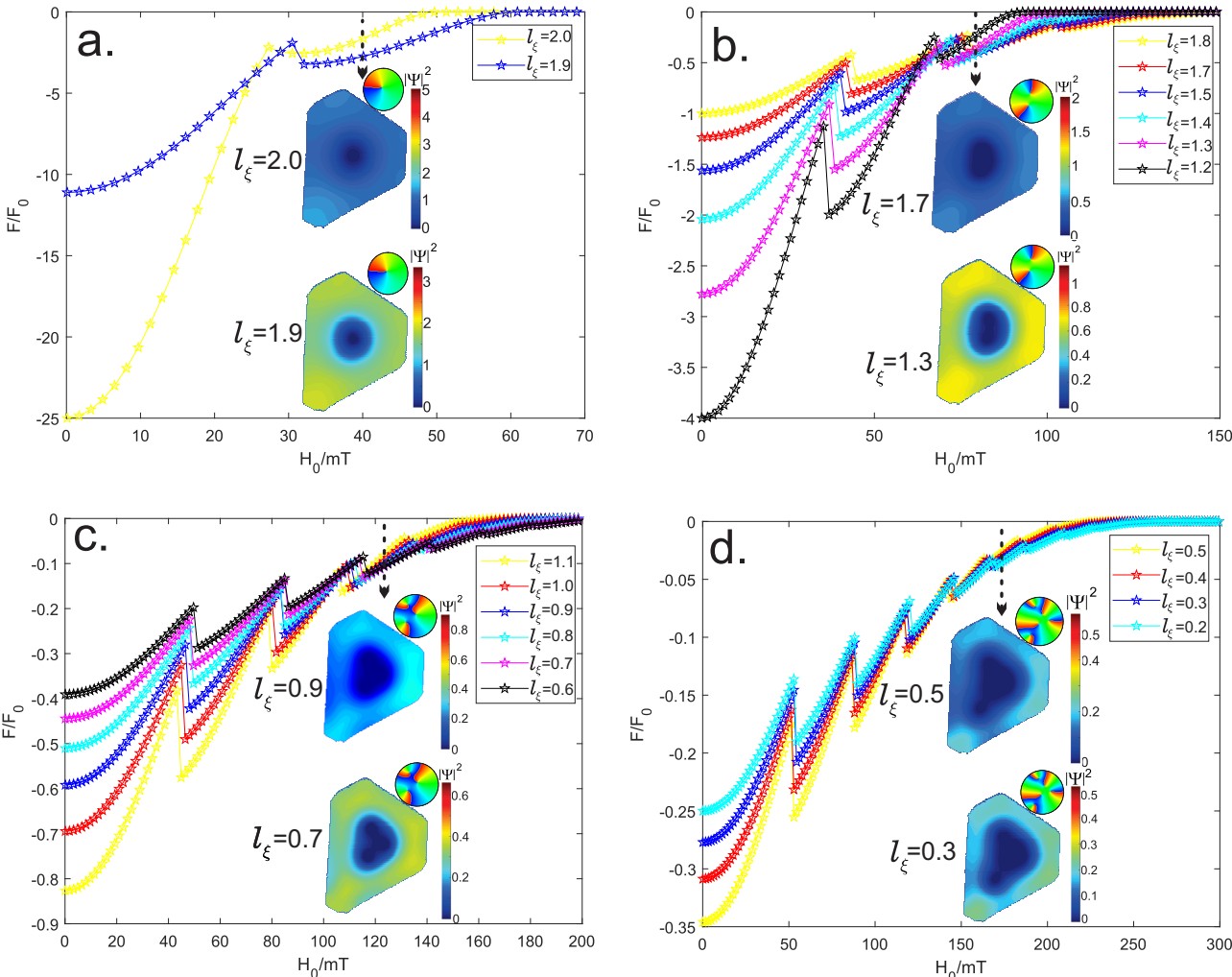

**Figure 3.** (Color online) Gibbs free energy curves for the island with different values of *l* as a function of the magnetic field applied perpendicular to the sample. Selected snapshots of the order parameter density are shown inserted below the free energy curves and the number of vortices within the island void are determined by the circular figures at the top, by calculating the phase of the order parameter for the values of the applied magnetic field $H_0$ demarcated by the arrow in figures a., b., c., and d.

In order to identify the observed quantum phenomenon, we explored the evolution and stability of the superconductivity in the islands with the magnetic field (Figures 2–4). At zero magnetic field, Figure 2 snapshot (a.) shows a spatially homogeneous superconducting condensate to exist in the entire nanoisland, showing the Meissner ($L = 0$) state. As the field increases further, the snapshots in Figures 2 and 3 reveal novel intriguing vortex configurations, evolving until the normal state is achieved in each superconducting island, but in order to obtain a deeper insight into the field evolution of the condensate confined we need to focus on their phases.

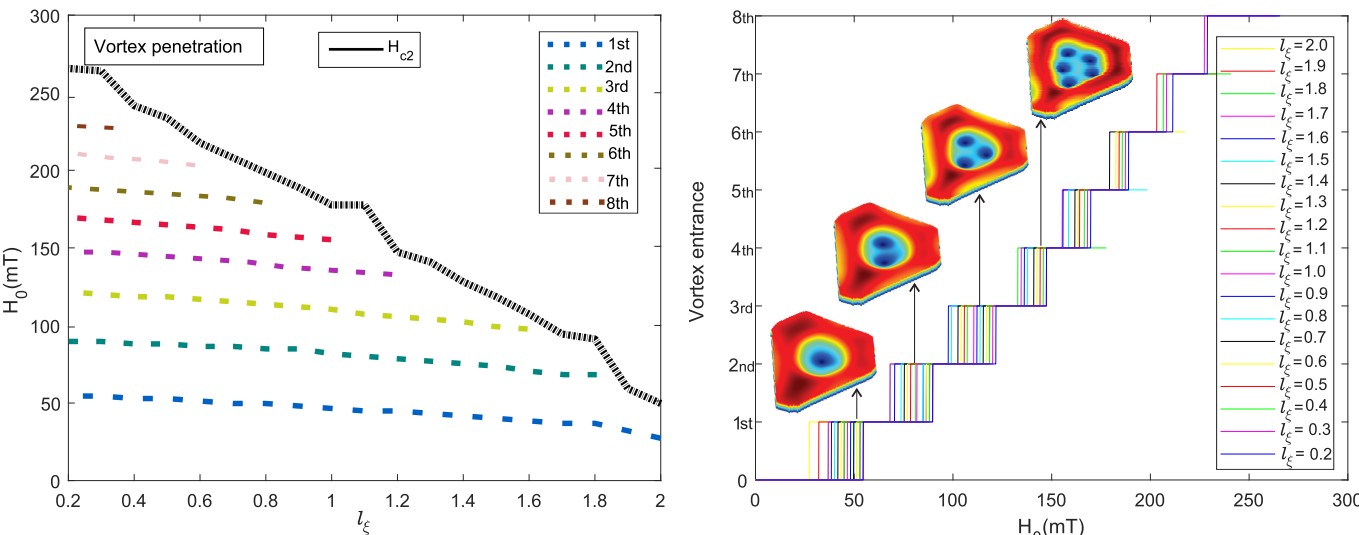

**Figure 4.** (Color online) (**Left**) Values of the applied magnetic field $H_0$ versus electron mean free path $l$, the green curve shows the values of the second critical magnetic field $H_2$ reached for each value of $l$ selected, showing the direct proportionality between both magnitudes. While the yellow curve shows the values of $H_0$ for which the Bean-Livistong barrier (BLB) [41] is exceeded $H_1$ (first critical field) or first entry of vortices, whose inclination from $l = 0.2$ to $l = 2.0$ of the BLB because in the first case, the sample is in the dirty limit ($l \ll \xi$), while in the second case, it is in the clean limit ($l \gg \xi$). (**Right**) Entry of vortices in the sample as a function of $H_0$, inset some configurations of vortices that show that up to approximate values of $H_0 = 150mT$ the vortices penetrate one by one. Each line in this graph has a correspondence in Figure 3, where each jump in the Gibbs free energy curve demonstrates the entry of vortices.

In Figure 5, we present a color-coded diagram of (ZBC) vs. a magnetic field taken over a line crossing the island (depicted as a black line in the blue frame). This diagram shows a series of abrupt steplike transitions which are identified in separating the states with different vorticity $L$, where the colors from red (superconducting state) to blue (normal state) correspond to the local strength of superconducting condensate. Precisely, until $H_0 \approx 27.5$ mT the island remain in the Meissner state ($L = 0$) for $l = 2.0$, in agreement with the color map in Figure 5. Also, we can notice that only one vortex penetrates due to only one steplike transition taking place for $l = 2.0$ until it reaches the total normal state at $H_0 \approx 46.49$ mT (all regions became blue). In order to obtain a deeper insight into these phases, we focused on the field evolution of the condensate confined in the superconducting island with $l = 0.8$. At $H_0 \approx 46.49$ mT T the first vortex appears in the sample; it is clearly identified by its normal state core (blue region). This single vortex state lasts until $H_0 \approx 78.49$ mT. The $L = 2$ phase occurs at $H_0 \approx 78.49$ mT $< H_0 < H_0 \approx 110.49$ mT, followed by the $L = 3$ state, which also can be noticed in the CPD in the red snapshot located up of the $H_0 \approx 110.49$ mT $< H_0 < H_0 \approx 134.49$ mT.

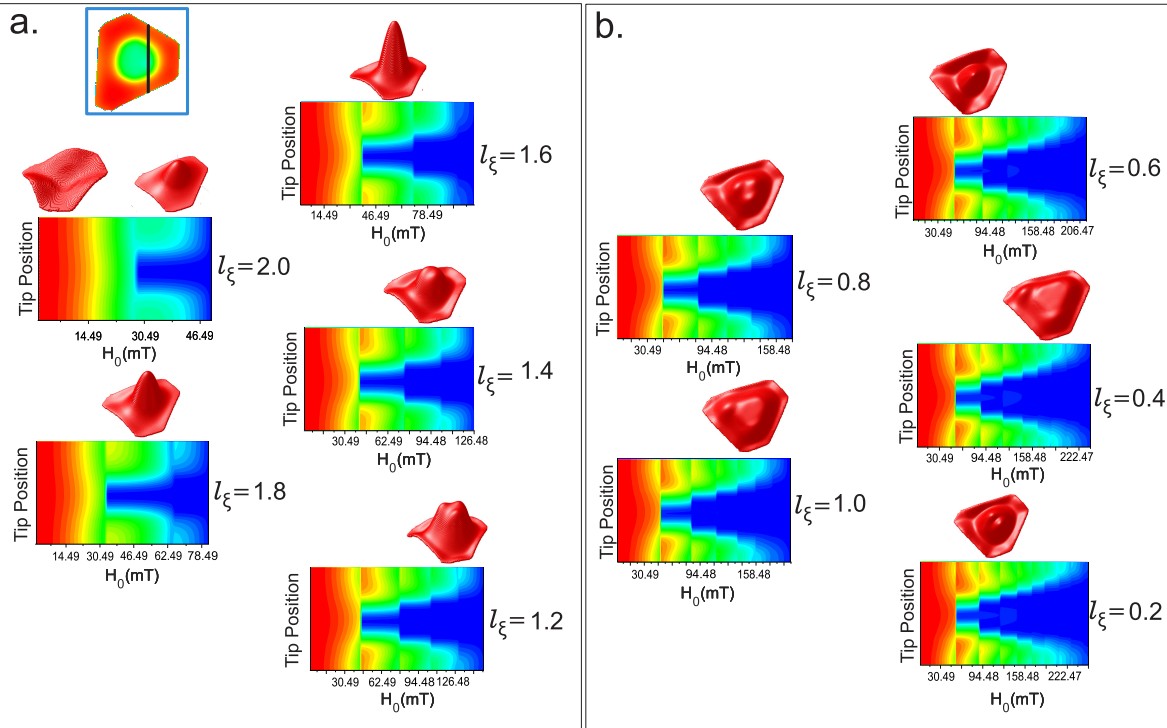

**Figure 5.** (Color online) (**a**,**b**) Phase diagrams of the value of Cooper-pair density (CPD) under the tip as a function of a magnetic field, obtained as the tip is moved along the line that runs through the island, and taken during the sweep up and down of the magnetic field Where CPD acquires the highest values in the red areas while the lowest values are demarcated with blue. The corresponding spatial distribution of CDP (red snapshot) for selected steplike transitions where every peak corresponds to the vortex position in the island. The black line in the figure inside the blue frame indicates the trajectory over which the tip was moved in order to simulate the scanning tunneling microscope (STM) to obtain the zero-bias conductance (ZBC) phase diagram.

Remarkably, in $L = 2$ for $l = 1.4$ looks like a single round object located at the island center, instead of two individual vortices. The straightforward conclusion is that here one (two) individual vortex cores are merged to form a single giant vortex.

## 3. Materials and Methods

We simulated a superconducting Pb island grown on a silicon substrate (111) as illustrated in Figure 1, but also, the thickness change $d$ and the mean free path $l_e$. The framework for our theoretical studies is the phenomenological Ginzburg–Landau (GL) theory [1]. We used the expressions for GL coefficients $\alpha$ and $\beta$ in the dirty limit, to include the variation $l_e$ the electron mean free path in the sample, i.e.,

$$\alpha(T) = -1.36 \frac{\hbar}{2m * \xi_0 l_\xi} \left(1 - \frac{T}{T_c}\right) = \frac{\alpha_0}{l_{\xi_0}} \left(1 - \frac{T}{T_c}\right) \tag{1}$$

$$\beta = \frac{0.2}{N(0)} \left(\frac{\hbar^2}{2m * \xi_0 l_e k_B T_c}\right)^2 = \frac{\beta_0}{l_{\xi_0}^2} \tag{2}$$

where $l_\xi = \frac{l_e}{\xi_0} \approx 2d(x,y,z)$ is the ratio of the electron mean free path ($l_e$) and BCS coherence length ($\xi_0$), whereas $d(x,y,z)$ is the thickness. The dimensionless form of the GL equation can be written as follow:

$$(-i\nabla - \mathbf{A})^2 \Psi = \left( \frac{1.367}{2d(x,y,z)} - \frac{1}{(2d(x,y,z))^2}|\Psi|^2 \right) \Psi \qquad (3)$$

where lengths are scaled to $\xi_0$, penetration depth $\lambda_0$ is defined as $\lambda_0^2 = mc^2\beta_0/16\pi|a_0|e^2$, the vector potential $\mathbf{A}$ is expressed $\phi_0/2\pi\xi_0$, and the order parameter is in units of $\Psi_0 = \sqrt{-\alpha/\beta}$. In this case, the sample has non-uniform thickness $d(x,y)$, but is sufficiently thin to assume the constant order parameter in the z-direction and solve only the first GL equation, Equation (3) has to be expanded on the left side by the term $i\frac{\nabla d(x,y)}{d(x,y)}(-i\nabla - \mathbf{A})$, as derived and used in Ref. [42–44]. Finally, we obtain our equation to be solved:

$$\begin{aligned} (-i\nabla - \mathbf{A})^2 \Psi = & \left( \frac{1.367}{(2d(x,y,z))} - \frac{1}{(2d(x,y,z))^2}|\Psi|^2 \right) \Psi \\ & - i\frac{\nabla d(x,y,z)}{d(x,y,z)} \cdot (-i\nabla - \mathbf{A})\Psi \end{aligned} \qquad (4)$$

## 4. Conclusions

In conclusion, strong vortex confinement effects were studied on a Pb island grown on Si (111), taking into account several changes of electron mean-free path, considering that experimental ion bombardment can be performed over a superconducting sample, by using the expressions for Ginzburg–Landau (GL) coefficients $\alpha$ and $\beta$ in the dirty limit. In these type II superconducting islands, an unexpected behavior of vortex configuration was studied, but also, their critical parameters allow control of this heterostructure in order to be applied in electronic devices.

**Author Contributions:** Derived G-L equations for dirty limit and performed the simulations, J.G. (Jesús González) and J.G. (Jader González); prepared the figures, F.D. and C.S.; software, J.G. (Jesús González); formal analysis, J.G. (Jesús González) and J.G. (Jader González); writing original draft preparation, J.G. (Jorge Gómez) and J.G. (Jesús González). All authors have read and agreed to the published version of the manuscript.

**Funding:** This research was funded by the Universidad del Magdalena (Fonciencias) and Dirección de Investigación (Colombia) y Transferencia (DIT) Universidad Pontificia Bolivariana (Colombia).

**Data Availability Statement:** Not applicable.

**Acknowledgments:** The authors appreciate the support of the Universidad del Magdalena (Fonciencias) and Dirección de Investigación (Colombia) y Transferencia (DIT) Universidad Pontificia Bolivariana (Colombia).

**Conflicts of Interest:** The authors declare no conflict of interest. The founding sponsors had no role in the design of the study; in the collection, analyses, or interpretation of data; in the writing of the manuscript, and in the decision to publish the results.

## Appendix A

**Table A1.** Stability range of vortex states for each value of the mean free path studied. Second critical field is included in the last column.

| l/nm | 1st/mT | 2nd/mT | 3rd/mT | 4th/mT | 5th/mT | 6th/mT | 7th/mT | 8th/mT | $H_2$/mT |
|------|--------|--------|--------|--------|--------|--------|--------|--------|----------|
| 0.2  | 27.29  |        |        |        |        |        |        |        | 49.69    |
| 0.3  | 32.09  |        |        |        |        |        |        |        | 59.29    |
| 0.4  | 36.89  | 68.89  |        |        |        |        |        |        | 91.28    |
| 0.5  | 36.89  | 68.29  |        |        |        |        |        |        | 94.48    |
| 0.6  | 38.49  | 70.49  | 97.68  |        |        |        |        |        | 107.28   |
| 0.7  | 40.09  | 73.69  | 99.28  |        |        |        |        |        | 118.48   |
| 0.8  | 41.69  | 75.29  | 102.48 |        |        |        |        |        | 128.08   |
| 0.9  | 43.29  | 76.89  | 104.08 |        |        |        |        |        | 140.88   |
| 1.0  | 44.89  | 78.49  | 105.68 |        |        |        |        |        | 147.28   |
| 1.1  | 44.89  | 80.09  | 107.28 | 132.88 |        |        |        |        | 177.67   |
| 1.2  | 46.49  | 81.68  | 110.48 | 134.48 | 155.28 |        |        |        | 177.67   |
| 1.3  | 48.09  | 84.88  | 112.08 | 136.08 | 156.88 |        |        |        | 188.87   |
| 1.4  | 49.69  | 84.88  | 113.68 | 137.68 | 158.48 |        |        |        | 198.47   |
| 1.5  | 49.69  | 86.48  | 115.28 | 140.88 | 161.68 | 179.27 |        |        | 208.07   |
| 1.6  | 51.29  | 86.48  | 116.88 | 142.48 | 163.28 | 182.47 |        |        | 217.67   |
| 1.7  | 52.89  | 88.08  | 118.48 | 144.08 | 164.87 | 184.07 | 203.27 |        | 233.67   |
| 1.8  | 52.89  | 88.08  | 118.48 | 145.68 | 166.47 | 185.67 | 206.47 |        | 241.67   |
| 1.9  | 54.49  | 89.68  | 120.08 | 147.28 | 168.07 | 187.27 | 208.07 | 227.27 | 264.06   |
| 2.0  | 54.49  | 89.68  | 121.68 | 147.27 | 169.67 | 188.87 | 211.27 | 228.80 | 265.66   |

## Appendix B

Figure A1 shows the modification of the mean free path in selected zones of Pb island, which is totally possible using nanopattering Ion technique [29–31]. The colors of the curves in the figure are linked to the frames of the inserted snapshots. Where we can see that the vortices enter from the top, where the mean free path electron (*l*) is smaller and in turn the superconducting condensate is smaller according to the color bar. Additionally, it is observed that the normal state is reached first in the lower part where *l* average is greater. All of the above demonstrates the relationship of the mean free path electron with the second critical field ($H_2$), where $H_2 \propto 1/\xi(0)$ and $\xi \propto \sqrt{l}$, which increases with the reduction of *l* while this decreases with the increase of *l*. Additional simulations are shown in the lower panels, where panel a. shows what happened to the superconducting island by now decreasing the *l* at the bottom of the sample, while in panel b. the reduction of *l* occurs on the right side, the opposite case is shown in panel c. The panels a. b. and c. correctly simulate what was previously explained.

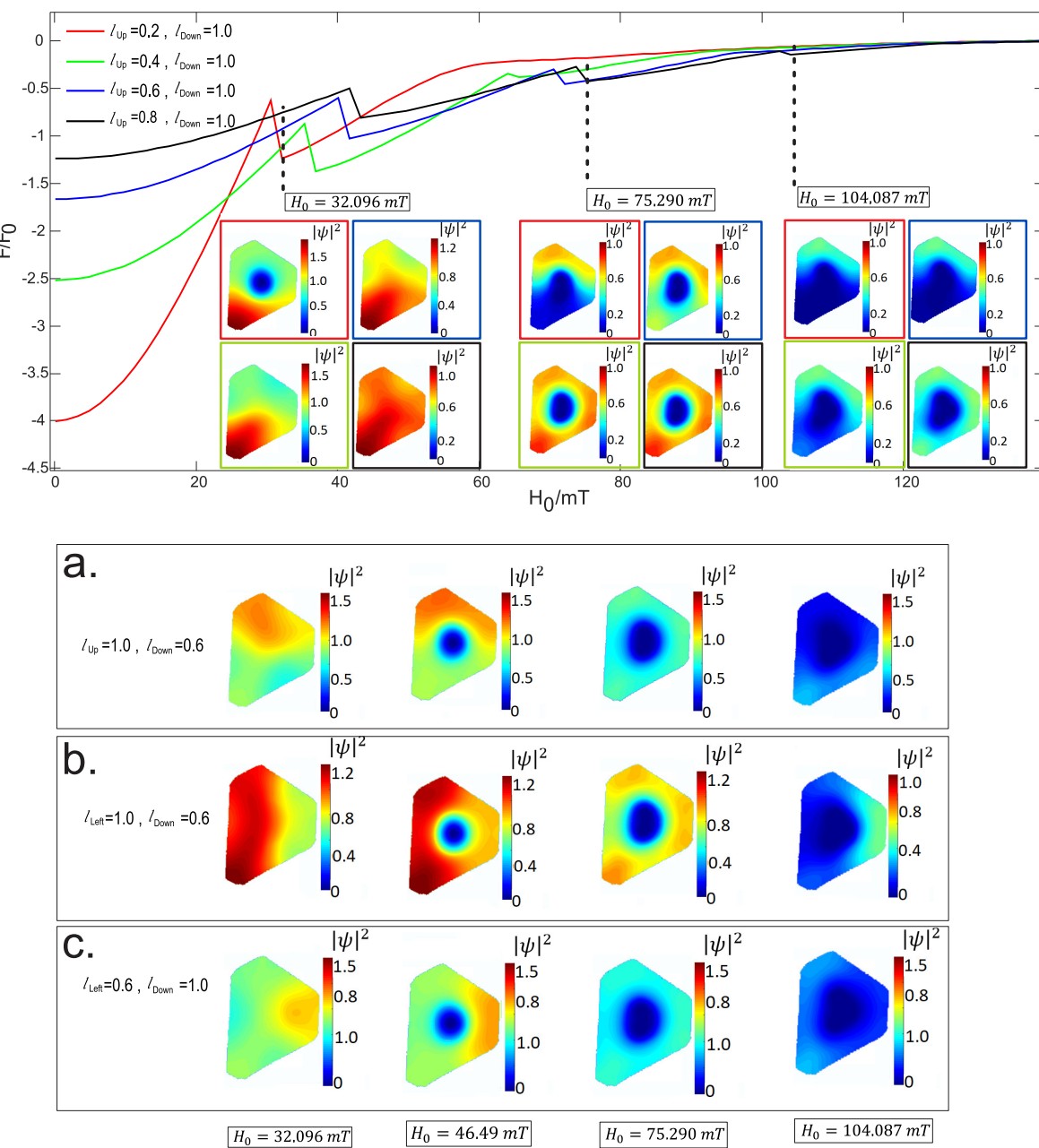

**Figure A1.** The figure shows the free energy as a function of the applied magnetic field. $l_{up}$ and $l_{down}$ show the values of mean free path at the top and bottom, respectively, while $l_{left}$ shows $l_{rigth}$ for the left and right zones. Inside the figure we can see the clear modification of the vortex input under the influence of the mean free path, which implies a greater control of the superconducting condensate and the vortex configuration.

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
