# Peer review of "Effect of the Spatially-Varied Electron Mean Free Path on Vortex Matter in a Superconducting Pb Island Grown on Si (111)"

_condensedmatter, doi:10.3390/condmat8030077_

Round 1

Reviewer 1 Report

In the present work, vortices in superconducting islands are investigated theoretically using the Ginzburg-Landau equation. The study focuses on the case of strong confinement, when the size of an island is small with respect to the effective London penetration depth.

The energy and configurations of vortex states have been calculated as functions of the applied magnetic field. They represent an interest in the context of possible links to experimental observations of vortices in superconducting islands in the strong confinement regime.

The paper is clearly written and contains interesting results. In my opinion, it deserves publication in Condensed Matter after a careful check of references: for example, the correct page number in Ref. 2 should be 11793 instead of 173.

Author Response

Response to Reviewer 1 Comments

The authors want to thank the reviewer for his valuable comments and his words towards our research work.

Point 1: The paper is clearly written and contains interesting results. In my opinion, it deserves publication in Condensed Matter after a careful check of references: for example, the correct page number in Ref. 2 should be 11793 instead of 173.

Response: The references were checked in detail and the one corresponding to the reviewer's suggestion was corrected.

Reviewer 2 Report

Congratulations on your successful and interesting work. I read it with interest. I wish you further success in your scientific work. 

No comments.

Author Response

The authors want to thank the reviewer for his valuable comments and his words towards our research work.

Reviewer 3 Report

Authors have studied theoretically magnetic properties of small size superconductor. There are many works on this subject, some of them are cited in the present paper. The most interesting and strange to me result is increase of Hc2 with increasing mean path length \ell (see Fig. 5), but it is not discussed in the paper. Indeed, in dirty limit Hc2~1/\ell and it should increase, but authors find opposite behavior.  May be the reason for this result is in details of used model? By the way, to me it is not clear how authors incorporated variation of thickness in the model. They also say nothing how they numerically solve GL equaitons and did they solve or not Maxwell equation for vector potential?

I think in the present form paper cannot be published.  Authors should present details of their model and dicsuss why in their model Hc2 increases with increase of \ell which contradicts textbooks on superconductivity. Other results are not interesting, they resemble previous works. 

English is poor. I can understand what authors want to say but it takes some efforts from me.  

Author Response

Response to Reviewer 3 Comments

Point 1: The most interesting and strange to me result is increase of Hc2 with increasing mean path length \ell (see Fig. 5), but it is not discussed in the paper. Indeed, in dirty limit Hc2~1/\ell and it should increase, but authors find opposite behavior. May be the reason for this result is in details of used model? 

Response: The authors would like to thank the reviewer for pointing out his comments in Figure 5. After an extensive review of the simulation data, we see that the error was made in the graph when placing the electron mean free path values. All simulation results were validated and discussed with fellow superconductivity experts.

Figure 5. was corrected, as well as the data in the discussion about it. Lines 151-161. Additionally, with the objective of expanding the behavior of the second critical field H2 with respect to the modification of the mean free path of the electron, a paragraph was included in appendix B. (lines 190-202)

Point 2: By the way, to me it is not clear how authors incorporated variation of thickness in the model. They also say nothing how they numerically solve GL equaitons and did they solve or not Maxwell equation for vector potential?

Response: Lines 61-64 explain the inclusion of the thickness, as well as the inclusion of the references where the numerical method is explained. The latter was not included but referenced so as not to extend the paper with mathematical and numerical details.

Point 3: Comments on the Quality of English Language. English is poor. I can understand what authors want to say but it takes some efforts from me.  

Response: We used the service of English Language Editing by MDPI, but only few mistakes were found.

Round 2

Reviewer 3 Report

Authors mainly answered about their model but I still have a question about \ell. Does it vary across the sample or not, like thickness? If yes then it should be additional term in GL equation.

Still there is a question about dependence Hc2 on \ell. In Fig. 4 Hc2 increases with increase of \ell which contradicts to textbooks on superconductivity where Hc2 ~ 1/\ell. Is it correct result on Fig. 4 or there is some mistake? If it is correct then authors should discuss it in details because it is the main and new result, the other results are trivial.

Author Response

Response to Reviewer 3 Comments

The authors would like to thank the reviewer for the essential contributions made to our work which offers us the opportunity to gain experience in the dissemination of the theoretical study of superconducting systems.

Point 1: Authors mainly answered about their model but I still have a question about \ell. Does it vary across the sample or not, like thickness? If yes then it should be additional term in GL equation.

Yes, it varies on the sample, this is the foundation of our work. As detailed in the introduction, it is possible to change \ell at will with experimental methods. Even universities close to ours can do this ion implantation and create the patterns in which this inclusion of impurities varies, which would modify the mean free path of the electron. . The variation of \ell in the mesh of the numerical solution is in the term on the right of Eq.3. It is possible to include the relationship of thickness with \ell, even with critical temperature, but we have not yet fully derived the G-L equations for the inclusion of these additional effects, if this is the direction your question is pointing.

Point 2: Still there is a question about dependence Hc2 on \ell. In Fig. 4 Hc2 increases with increase of \ell which contradicts to textbooks on superconductivity where Hc2 ~ 1/\ell. Is it correct result on Fig. 4 or there is some mistake? If it is correct then authors should discuss it in details because it is the main and new result, the other results are trivial.

The results of Fig.4 (left) were detailed and corrected. Fig. 4 (right) is in agreement with the relation of the number of vortices y \ell for the different values of the applied magnetic field.
